# Maternal Diet, Infection, and Risk of Cord Blood Inflammation in the Bangladesh Projahnmo Pregnancy Cohort

**DOI:** 10.3390/nu13113792

**Published:** 2021-10-26

**Authors:** Anne CC Lee, Sara Cherkerzian, Ingrid E Olson, Salahuddin Ahmed, Nabidul Haque Chowdhury, Rasheda Khanam, Sayedur Rahman, Chloe Andrews, Abdullah H Baqui, Wafaie Fawzi, Terrie E Inder, Stephanie Nartey, Charles A Nelson, Emily Oken, Sarbattama Sen, Raina Fichorova

**Affiliations:** 1Department of Pediatric Newborn Medicine, Brigham and Women’s Hospital, Boston, MA 02115, USA; scherkerzian@bwh.harvard.edu (S.C.); iolson@bwh.harvard.edu (I.E.O.); candrews@bwh.harvard.edu (C.A.); tinder@bwh.harvard.edu (T.E.I.); ssen2@bwh.harvard.edu (S.S.); 2Harvard Medical School, Boston, MA 02115, USA; Charles.Nelson@childrens.harvard.edu (C.A.N.); emily_oken@harvardpilgrim.org (E.O.); rfichorova@bwh.harvard.edu (R.F.); 3Projahnmo Research Foundation, Banani, Dhaka 1213, Bangladesh; sahmed@prfbd.org (S.A.); nchowdhury@prfbd.org (N.H.C.); srahman@prfbd.org (S.R.); 4Johns Hopkins Bloomberg School of Public Health, Baltimore, MD 21205, USA; rkhanam1@jhu.edu (R.K.); abaqui@jhu.edu (A.H.B.); 5Harvard T.H. Chan School of Public Health, Boston, MA 02115, USA; mina@hsph.harvard.edu; 6Department of Obstetrics, Gynecology and Reproductive Biology, Brigham and Women’s Hospital, Boston, MA 02115, USA; snartey@bwh.harvard.edu; 7Boston Children’s Hospital, Boston, MA 02115, USA; 8Harvard Graduate School of Education, Boston, MA 02138, USA; 9Harvard Pilgrim Health Care Institute, Boston, MA 02215, USA

**Keywords:** undernutrition, prenatal infection, inflammation, micronutrient

## Abstract

Inflammation may adversely affect early human brain development. We aimed to assess the role of maternal nutrition and infections on cord blood inflammation. In a pregnancy cohort in Sylhet, Bangladesh, we enrolled 251 consecutive pregnancies resulting in a term livebirth from July 2016–March 2017. Stillbirths, preterm births, and cases of neonatal encephalopathy were excluded. We prospectively collected data on maternal diet (food frequency questionnaire) and morbidity, and analyzed umbilical cord blood for interleukin (IL)-1α, IL-1β, IL-6, IL-8 and C-reactive protein. We determined associations between nutrition and infection exposures and cord cytokine elevation (≥75% vs. <75%) using logistic regression, adjusting for confounders. One-third of mothers were underweight (BMI < 18.5 kg/m^2^) at enrollment. Antenatal and intrapartum infections were observed among 4.8% and 15.9% of the sample, respectively. Low pregnancy intakes of B vitamins (B1, B2, B3, B6, B9 (folate)), fat-soluble vitamins (D, E), iron, zinc, and linoleic acid (lowest vs. middle tertile) were associated with higher risk of inflammation, particularly IL-8. There was a non-significant trend of increased risk of IL-8 and IL-6 elevation with history of ante-and intrapartum infections, respectively. In Bangladesh, improving micronutrient intake and preventing pregnancy infections are targets to reduce fetal systemic inflammation and associated adverse neurodevelopmental outcomes.

## 1. Introduction

Globally, an estimated 200 million children under the age of 5 years do not reach their full potential in cognitive development, the vast majority in low-middle-income countries (LMIC) [1]. The first 1000 days from conception onward presents a critical window of opportunity to optimize child neurodevelopment [2]. Maternal undernutrition and infections are prevalent and targetable pregnancy risks in LMIC may have significant consequences for long term offspring development. There is increasing recognition of the importance of the complex “collision” of nutrition and inflammation in global child health, as evidenced by the National Institute of Child Health and Human Development INSPIRE (Inflammation and Nutritional Science for Programs/Policies and Interpretation of Research Evidence) initiative [3]. The relationships between maternal nutrition, infection, and maternal–fetal inflammation in pregnancy are complex and inadequately characterized, particularly in LMICs [4].

Maternal infections are prevalent in LMICs and associated with maternal and fetal systemic inflammation. The risk for maternal infections is high in LMIC, where prenatal care coverage is low, access to clean water and sanitation is poor, and up to half of births occur at home without a skilled birth attendant [5]. Up to 60% of mothers may experience a urinary or reproductive tract infection at least once in pregnancy [6,7], and in LMIC, screening and treatment of these infections is inadequate. Maternal infections result in inflammation by direct bacterial cytotoxicity and activation of host immune responses [8]. In the Extremely Low Gestational Age Newborn (ELGAN) study, pregnancy history of cervical/vaginal and urinary tract infections was associated with elevations in newborn pro-inflammatory cytokines [9]. Chorioamnionitis has been associated with elevations in maternal [10,11] and umbilical cord blood [12] cytokines.

Undernutrition is common among women of reproductive age in South Asia (21% underweight (BMI < 18.5 kg/m^2^)) [13] and linked with immune dysfunction. Chronic protein energy malnutrition impairs antigen-presenting cell and cell mediated T-cell function, thereby increasing infection risk [3,14,15]. Furthermore, chronic undernutrition may also activate the hypothalamic-pituitary-adrenal axis and result in immune dysregulation [16]. In rural Bangladesh, 63% of pregnant mothers have at least one micronutrient deficiency [17]. Specific micronutrients such as long-chain polyunsaturated fatty acids (PUFAs), B vitamins, and zinc, modulate immune function and the inflammatory response, and thus, deficiencies may result in a pro-inflammatory milieu. Finally, maternal macro- and micronutrients may be epigenetic regulators of inflammatory and immune function in the fetus [18].

Fetal and neonatal inflammation leads to white matter brain injury and has been linked with altered neonatal functional connectivity [19,20], abnormal behavior, and autism [21]. In preterm infants, perinatal inflammation has been associated with ventriculomegaly, microcephaly and lower neurodevelopmental scores as well as intelligence quotient in childhood [22,23]. In order to identify potential modifiable pregnancy targets to reduce fetal exposure to inflammation, we aimed to investigate the association of maternal pregnancy undernutrition and infections with offspring inflammation, as measured by pro-inflammatory immune mediators in umbilical cord blood in a pregnancy-birth cohort in rural Bangladesh.

## 2. Materials and Methods

This study was implemented in the Projahnmo field site, a site for maternal, newborn, and child health research, which was established in 2001 in Sylhet district of Bangladesh by a partnership of Johns Hopkins University, the Bangladesh Ministry of Health and Family Welfare (MOHFW), and several Bangladeshi NGOs and academia. The Projahnmo study site is located in two sub-districts of Sylhet district, in northeast Bangladesh (Kanaighat and Zakiganj: population: ~500,000), ~350 km from the capital city, Dhaka. Since its inception, Projahnmo has maintained routine demographic surveillance for all pregnancy and births in the study areas. The study population is rural, poor (average daily wage < USD 2/day), and with high disease burden. Mortality rates are high (stillbirth rate 30/1000 births, neonatal mortality rate 37/1000 live births) [24]. In this study, in 299 consecutively enrolled women in the Projahnmo pregnancy cohort, we collected umbilical cord blood spots at delivery between July 2016–March 2017. In the present analysis, we included pregnancies in this cohort resulting in a full-term live birth. We excluded stillbirths, infants with neonatal encephalopathy, and preterm births (*n* = 48 total), yielding 251 pregnancies for analysis.

Trained community health workers (CHWs) collected primary data and measurements at home visits on household sociodemographic, environmental and pregnancy risks, medical and obstetric histories, pregnancy outcomes and maternal-newborn anthropometrics and morbidity. A table of data collection timepoints can be viewed in Appendix A. Birth attendants were interviewed by research staff regarding delivery history and maternal/infant morbidity, including neonatal encephalopathy and maternal puerperal infections/sepsis. CHWs followed over 90% of pregnancies in the population after delivery and conducted postnatal visits within 1 to 6 days and 42–60 days postpartum. This study was approved by the Ethics Review Committee of Johns Hopkins University (IRB no. 00004508), International Centre for Diarrhoeal Disease Research Bangladesh (icddr,b) in Bangladesh (PR 12073) and Partners Health Care (Protocol 2014P001741).

### 2.1. Exposure Measures

#### 2.1.1. Maternal Nutritional Status

CHWs measured maternal anthropometrics at the enrollment (<20 weeks gestation), 24–28 week, 32–36 week, and 38–40 week antenatal care (ANC) visits. Weight was measured with OMRON Digital Body Weight Scale (model: HN-283, 100 g precision) that was calibrated daily. Height was measured with a locally constructed portable height stadiometer (precision: 1 mm) and mid-upper arm circumference (MUAC) using a TALC insertion tape (precision: 1 mm). Training and standardization in anthropometric measurements were performed by research staff with anthropometry expertise using a protocol modified from the WHO Multicentre Growth Reference Study [25], at the start of the study and every 3 months thereafter. All measures were performed independently three times, and 10% of measures were further repeated by a field supervisor blinded to the CHWs measures. Maternal body mass index (BMI) was calculated as weight in kg divided by height in meters squared at the first ANC study visit (mean gestational age at enrollment 12.1 weeks (standard deviation(SD) 3.8 weeks)). Maternal underweight was classified as BMI < 18.5 kg/m^2^ and overweight or obese was classified as BMI ≥ 25 kg/m^2^. Low MUAC was classified as MUAC < 22 cm and stunting was classified as height <145 cm. Anemia was classified as maternal enrollment hemoglobin (Hb) levels <11 g/dL.

#### 2.1.2. Micronutrient Intake

Maternal dietary intake was assessed using a thirty-nine-item food frequency questionnaire collected during ANC visits at 24–28, 32–36, and 38–40 weeks’ gestation. Data collectors interviewed women on their recall of food item frequency for the last 7 days (ranked categorically: never, once per week, 2–4 times per week, 5–6 times per week, and daily). Food items from the food frequency questionnaire (FFQ) were matched with corresponding food items and their nutrient data provided by Food Composition Tables for Bangladesh [26]. For nutrients without adequate data available in Bangladesh tables, we used the Indian Food Composition Tables [27]. Nutrient data for FFQ variables containing more than one possible food item (i.e., “any red meat”) were averaged across all possible items (i.e., beef, goat, etc.). Portion sizes were not collected with the original questionnaire and thus we used the recommended daily serving sizes from the Dietary Guidelines for Bangladesh [28] to estimate nutrient intakes. Daily nutrient intake was estimated as the nutrient value * recommended daily serving/portion size (g) * daily consumption frequency per maternal self-report. Dietary nutrient intake, based on average consumption frequency of the 24–28 and 32–36 FFQ assessments to characterize habitual pregnancy intake prior to delivery, was categorized into tertiles for analysis.

#### 2.1.3. Maternal Infections

CHWs interviewed mothers and birth attendants regarding history of intrapartum fever, clinical chorioamnionitis, duration of rupture of membranes, administration of antibiotics during delivery, and report of infections during current pregnancy including: UTI, sexually transmitted diseases, and other reproductive tract and systemic infections. Infection exposure was examined by timing of exposure: during pregnancy (24–28 week, 32–36 week, or 38–40 week), delivery, as well as a binary variable indicating any maternal infection at any time.

### 2.2. Outcome Measures

#### 2.2.1. Blood Spot Collection

Three drops of umbilical cord blood (0.25–0.20 mL) were collected on Whatman filter paper (FTA (Flinders Technology Associates) Card, General Electric) at delivery by research staff. Blood spots were air-dried at room temperature for at least 24 h, and stored in sealed Ziplock bags with desiccant at −80 °C.

#### 2.2.2. Inflammatory Protein Analysis

Samples were sent to the Laboratory of Genital Tract Biology, Brigham and Women’s Hospital (Boston, MA, USA), which conducted the inflammation analysis for the ELGAN study [9,22,29]. Proteins were eluted from dried blood spots as per standard procedures developed and validated for the ELGAN study [30]. Inflammation biomarkers were selected based on analytic validation and prior data demonstrating the associations with our exposures of interest, as well as neurodevelopmental outcomes [31,32]. Cord blood levels of IL-1α, IL-1β, IL-6, IL-8, and CRP were measured using electrochemiluminescence multiplex assays on a Meso Scale Discovery Sector Imager S600 (MSD, Gaithersburg, Maryland). The platform has been analytically and clinically validated compared to traditional ELISA [23,29,33,34,35,36,37,38]. Protein concentrations were calculated from relative luminescent units based on interpolation from log calibrator curves. Reproducibility was confirmed by reanalyzing a quality control split sample on each assay plate, which showed an inter-assay variation of 5.54% for IL-1α, 5.96% for IL-1β, 8.44% for IL-6, 10.21% for IL-8 and 5.10% for CRP. Each analyte was normalized by the total protein content (mg) as determined by BCA assay. Similar methods have been used and reported in ELGAN studies [23,31,32].

### 2.3. Statistical Analysis

We summarized descriptive statistics of key demographic and clinical characteristics, exposure, and outcome variables. Weekly FFQ food item consumption at each time point of assessment was calculated as a weighted average (weekly frequency category value (0 (none), 1 (once), 3 (2–4 times), 5.5 (5–6 times), and 7 (daily)) * percent each frequency category was observed in sample) multiplied by the recommended serving size. The correlations among nutrient concentrations were determined by Spearman correlation coefficient and plotted on a heat map. The relationship between each exposure (maternal underweight, overweight/obese, low MUAC, stunting, anemia, nutrient intake (dietary intake tertiles of vitamins A, B (B1, B2, B3, B6, B9, B12), C; minerals (iron, selenium, zinc)); and long-chain polyunsaturated fatty acids (LCPUFAs) (alpha-linolenic acid, arachidonic acid, docosahexaenoic acid, eicosapentaenoic acid, eicosatrienoic acid, linoleic acid); or maternal infections in pregnancy (antepartum, intrapartum)) and outcome (elevated levels of inflammatory protein concentrations) was calculated using logistic regression. Concentrations of individual cytokines were categorized as ≥75% vs. <75% for analysis [29]. The statistical approach of using the top-quartile concentrations was supported by the concept that the highest concentrations would be most biologically significant and by the observed nonlinear distribution of the concentrations of most proteins. The approach has been validated by clinical content showing strong association between newborn inflammation defined as top quartile concentration of the inflammatory proteins chosen for this study, infections and child development outcomes [23,39]. Composite variables of exposure groups (fat-soluble or water-soluble vitamins, minerals, LCPUFAs) and outcomes (elevation of one or more inflammatory biomarkers) were also created. Multivariate models were constructed to adjust for potential confounding. Based on our a priori conceptual model, we selected the following potential confounders: socioeconomic status, nulliparity, MUAC > 22 cm, maternal educational attainment (years), tobacco and/or betel nut use, and season of initial antenatal assessment [40]. To categorize the seasons for multivariate analysis, due to small numbers, one subject in season category Hermanta was merged with Sarat and one subject in season category Shhit was merged with Basanta. We used principal component analysis to construct a wealth index from household possessions. For this exploratory analysis, adjustments were not made for multiple comparisons.

### 2.4. Statistical Power

We estimated the detectable effect size (odds ratio (OR)) given a two-sided test with alpha = 0.05 and power of at least 0.80 in a sample of *n* = 251 adjusted by a variance inflation factor (VIF) for model covariates ranging from 0 to 0.15. The detectable effect size ranged from adjusted OR (aOR) 2.14 to 2.28 [41].

## 3. Results

A total of 251 mothers and their full-term infants were included in the analysis; all were singleton births. Table 1 shows key demographic and clinical characteristics of the study population. Women were on average 23.7 years old (SD 4.6) with primary level education (mean years of schooling 6 (SD 2.9)). Rates of maternal undernutrition were high, 20.1% of women were stunted, 31.7% underweight at their first ANC/enrollment visit (mean GA 12.1 weeks), and 34.5% had a MUAC < 22 cm. Betel nut, the seed of the Areca palm tree, is a stimulant mixed with tobacco and chewed in this population, with 36.3% of pregnant women reporting use during the current pregnancy. All infants were born vaginally and 92.0% of deliveries in this sub-study occurred in a health facility. Mean birthweight was 2749.9 g (SD 413.7) and rates of term low birth weight (22.9%) and small for gestational age (43.9%) were high, similar to national rates in Bangladesh [42,43].

### 3.1. Maternal Dietary Intake

Maternal self-report of FFQ food item consumption is shown in Figure 1. The animal-source foods with the highest amount of average weekly consumption were fish (463–478 g/week) and poultry (80–108 g/week), followed by milk (77–92 g/week). Among plant-based food items, the foods with highest consumption were leafy green vegetables (324–362 g/week), green peppers (193–200 g/week), followed by mango (120–182 g/week). Maternal diet was relatively consistent in this population, with little variation across pregnancy gestation (Figure 1). Differences in mean food item intake across visits were of small magnitude (Appendix A) and significant for fewer than one-third of food items.

Table 2 shows the estimated nutrient intake of pregnant women in this population for 20 nutrients as determined from FFQ responses, compared to Institute of Medicine (IOM) recommended values in pregnancy [46]. Nutrient intake in this population was substantially lower than recommended intake in pregnancy for most nutrients [46], with the exception of Vitamin C. Figure 2 shows a correlation matrix of nutrient intake. The intakes of several vitamins were highly correlated (r > 0.80). Daily intake of most B vitamins (B3, B6 and B9) were highly correlated with each other, with relatively lower correlation with B12 intake. Vitamin B3, B6 and B9 intake was also strongly correlated with Vitamin D, Vitamin E, iron, and zinc intake. Correlation among the different LCPUFAs were also high.

### 3.2. Infection Prevalence

History of an antenatal infection was reported in 12 women (4.8%). Urinary tract infection was reported by 2 participants, and 10 women received antibiotics during the pregnancy to treat an infection (*n* = 10). Intrapartum infection was reported in 40 (15.9%) women, as indicated by fever during delivery (*n* = 6), receipt of antibiotics for presumed infection (*n* = 33), or abnormal vaginal discharge (*n* = 1) (self-reported or reported by a birth attendant). A total of 49 (19.5%) women reported infection at either time. There were inadequate numbers of infections to disaggregate the analysis by infection type.

#### 3.2.1. Distribution of Cord Blood Inflammation Biomarkers

The distributions of umbilical cord IL-1α, IL-1β, IL-6, IL-8, and CRP were highly right-skewed, as typical of inflammation biomarkers. Histograms of the natural log transformed (ln) distributions and thresholds for quartile selection are shown in Figure 3.

#### 3.2.2. Maternal Nutrition, Infection and Odds of Elevated Inflammatory Biomarkers

The odds ratios (ORs) and 95% confidence intervals for elevation (top quartile) of IL-1α, IL-1β, IL-6, IL-8, and CRP in umbilical cord blood are shown for maternal nutrition (anthropometrics and anemia) and infections in pregnancy (Figure 4). Maternal anemia (enrollment Hb level < 11 g/dL) was significantly associated with greater odds of elevated IL-1α (aOR = 1.92, 95% CI: 1.04, 3.54). No other statistically significant results (*p* < 0.05) were observed among these exposures, though ORs tended to be greater than one. Greater odds of elevated IL-6 had a borderline significant association with stunting (maternal height < 145 cm; aOR = 1.97, 95% CI: 0.97, 4.02) and intrapartum infections (aOR = 2.07, 95% CI: 0.95, 4.54). aORs for antenatal infection and for any antenatal or intrapartum infection, as well as for overweight BMI status (data not shown), were greater than one across all inflammatory markers, though none met statistical significance.

Table 3 shows the association between nutrient intake in pregnancy (tertiles) and cord blood cytokine elevation (reference group middle tertile (33rd–66th percentile) for all comparisons). Lowest tertile levels (<33rd percentile) of B vitamins (B1, B2, B3, B6, B9 (folate), B12) were associated with increased odds of inflammation, notably IL-1 α, IL-8 and IL-6, with aORs greater than 2 times the reference category (Table 3). In particular, low intake of Vitamin B2 was associated with significantly greater odds of elevation of all inflammatory markers with the exception of IL-1β. Low intake of iron and linoleic acid was associated with elevated IL-8, while low zinc and iron intake were associated with elevated IL-1α.

For Vitamins D and E, a relatively higher intake (>66%) was associated with lower aOR for IL-1α and IL-8 elevation, respectively. Higher B12 intake was associated with lower odds of elevated IL-6 and IL-8. As a group, low intake of fat-soluble vitamins (<33rd percentile intake of one or more of the fat-soluble vitamins (Vitamins A, D, and E)) was associated with greater odds of IL-8 elevation compared to the group without low intake of any fat-soluble vitamin. Low intake of any mineral (iron, selenium, and zinc) was associated with a greater odds of elevated levels of IL-1β and of IL-8 compared with those without a low mineral level.

## 4. Discussion

We report on pregnancy risk factors for cord blood inflammation in the Projahnmo birth cohort, a rural Bangladeshi population typical of South Asia with a high prevalence of maternal undernutrition. We found that lower intake of B vitamins (B1, B2, B3, B6, and B9 (folate)), iron, zinc, and omega-6 fatty acid (linoleic acid) in pregnancy, were associated with higher risk of inflammation (primarily IL-8) in the newborn at birth. Levels of fat-soluble vitamins D and E were also inversely associated with inflammation risk. Maternal stunting and infection at delivery were associated with a non-significant trend of increased risk of elevation of newborn IL-6.

While the link between diet and systemic inflammation is well established, fewer studies have examined this relationship in pregnancy, and particularly, on offspring inflammation. A recent systematic review summarized the evidence on associations between diet and maternal inflammation in pregnancy [47]. Despite inconsistent data, the authors reported that maternal diet with higher animal protein and cholesterol and lower fiber were associated with higher inflammation biomarkers (IL-6, IL-8, CRP or TNF-α) [47]. In Project Viva, a cohort of 1808 mother–child dyads in Massachusetts and a pro-inflammatory maternal diet (measured with the dietary inflammatory index (DII) [48], was positively associated with maternal systemic inflammation, measured by second trimester maternal serum CRP (β: 0.08 mg/L per 1-unit increase in maternal DII, 95% CI: 0.02, 0.14) [48]. Higher maternal DII has also been associated with higher maternal TNF-α [49] and IL-6 [50]. Greater fiber intake has been associated with lower CRP (2nd trimester) [51] and IL-8 (third trimester) [52] in a Danish pregnancy cohort. A majority of the data has been generated from high-income or Western settings which tended towards the pro-inflammatory state of excess fat/protein. Our study in Bangladesh is one of the first in a malnourished population where undernutrition and nutrient deficiencies are prevalent. Of note, in our cohort, mean intake of almost all nutrients was lower than recommended levels in pregnancy, and those nutrients associated with inflammation were those with greater degrees of deficiency. Furthermore, to the best of our knowledge, this is one of the first studies that has assessed the associations of maternal diet during pregnancy with inflammation in cord blood.

Low intake of B vitamins (B1, B2, B3, B6, and folate (B9)) were associated with cord blood inflammation. B vitamins have antioxidant properties and reduce oxidative stress, free radical generation, and regulate inflammatory cytokines [53,54,55,56,57,58]. B vitamins are also critical for energy production and DNA/RNA synthesis and repair [59]. Low riboflavin (B2) intake was associated with elevations of most of the inflammation proteins analyzed. B2 is a precursor or flavin adenine dinucleotide, that plays a central role in regulating oxidative stress and regulation of the inflammasome [60]. B2 deficiency has been associated with oxidant-mediated lung inflammation [61] and oral inflammation [62]. Vitamin B-6 has also been associated with inflammation (CRP) in the NHANES study (2003-2004) [63], and B-6 supplementation in rheumatoid arthritis patients has been shown to reduce levels of IL-6 and TNF-α [64]. We also found that higher B12 intake was associated with lower IL-6 and IL-8. A randomized controlled trial (RCT) of Vitamin B12 supplementation in pregnant Bangladeshi mothers resulted in lower risk of elevated AGP and CRP compared to the placebo arm [65]. Therefore, these data together suggest that B vitamins may play an important role in the modulation of inflammation in the maternal–fetal unit and an important intervention target in deficient populations.

Higher levels of intake of fat-soluble vitamins D and E were associated with reduced risk of cord inflammation in our Bangladeshi cohort. Vitamins D and E both have cell membrane antioxidant properties, inhibiting lipid peroxidation, mast cell activation and the expression of inflammatory cytokines [66], including IL-1 β, IL-6 and TNF- α. Vitamin D deficiency has been associated with elevated systemic proinflammatory cytokines in pregnancy and at delivery (IL-6 and TNF-α) [67] and low birth weight [68]. Our cohort carries higher risk of Vitamin D deficiency given the darker skin tone and clothing commonly covering exposed skin; other similar populations have demonstrated high rates of deficiency (~64%) [69]. An RCT of vitamin D supplementation (2000 IU/d) in pregnancy failed to demonstrate supplementation effects on inflammation, however in post hoc analysis, there was a negative association of serum 25(OH)D concentrations in late pregnancy and plasma concentrations of TNF-α, ICAM-1 and VCAM-1 [70]. Vitamin E deficiency has also been associated with inflammation, pre-eclampsia [71] and preterm birth, and small for gestational age. However, systematic reviews have failed to support the benefits of routine supplementation of prenatal Vitamin D and E on pregnancy outcomes [72,73] and routine supplementation is not currently recommended by WHO during ANC, though the American College of Obstetrics recommends consideration of Vitamin D deficiency screening in high-risk populations [74,75].

We observed increased inflammation risk among women in the lowest tertile of iron intake. Iron intake was highly correlated with the intake of several antioxidant vitamins (B, D and E) in our population. The association of low iron intake and inflammation may result from inflammation related to low intake of the B, D and E vitamins. Inflammation stimulates the master regulator of iron homeostasis, hepcidin, and decreases intestinal iron absorption [76,77]. Although we cannot determine causation in this observational study and we did not measure iron status or hepcidin to directly test this hypothesis, we postulate this relationship given the known pathways of iron homeostasis and the co-occurrence of B vitamins and iron in this cohort.

Zinc is an essential immune-modulatory micronutrient that targets nuclear factor kappa B (NF-kB), regulates pro-inflammatory responses and peroxisome proliferator activated receptor-α function [78], and plays a critical role in oxidative stress [79,80]. Maternal zinc deficiency has been associated with elevated maternal serum TNF-α and IL-8 and small-for-gestational-age births [81]. In a zinc supplementation RCT of neonates with clinical sepsis in India, zinc supplementation was found to reduce concentrations of serum TNF-α and IL-6 [82]. Our finding that low maternal zinc intake is associated with increased inflammation risk supports these clinical data, further emphasizing the role of zinc in healthy pregnancy outcomes.

We also found that higher intake of linoleic acid, an omega-6 LCPUFA, was associated with lower risk of newborn IL-6 and IL-8 elevation, respectively. While omega-6 LCPUFAs were previously thought to be pro-inflammatory, recent data implicate them in not only pro-inflammatory, but also in anti-inflammatory pathways [83,84,85], such as those related to prostacyclin and lipoxin A4. In an Italian cohort of 1123 adults, higher plasma omega-6 (arachidonic acid) was associated with lower IL-6 and IL-1 receptor antagonist [86]. Omega-6 LCPUFAs are associated with lower risk of cardiac events and now recommended at 5–10% dietary intake by the American Heart Association [85]. We did not see similar associations with omega-3 PUFAs in our cohort. While fish intake was high in this population, fish intake in this setting is different from typical western fish high in omega-3 LCPUFAs [87].

The cytokine that we consistently observed associations with maternal diet was IL-8. IL-8 (CXCL-8) is a chemokine that is a major effector of acute inflammation, attracting neutrophils and monocytes [88,89]. Unlike the IL-1 cytokines and IL-6 which are primary initiators of the inflammatory cascades leading to activation of proinflammatory transcription factors, IL-8 is a secondary mediator of inflammation, a downstream product of the cascades initiated by the primary cytokines or microbial products. Its expression is regulated by the synergism of several transcription factors including AP-1, NF-IL-6, and NF-kB [90]. Thus, higher levels of nutrients such as zinc that target transcription factors downstream from IL-1 and IL-6 may be reasonably expected to lower IL-8. Elevated IL-8 concentrations signify unopposed activation of the inflammatory cascade-and thus may be a better biomarker of sustained proinflammatory activation. We observed few significant associations with IL-1α, IL-1β, or CRP. In the recent systematic review by Yeh and colleagues, CRP was the most studied inflammation biomarker in pregnancy, however only 6 of 13 studies reported a significant relationship of CRP with diet. The cohort studied here included only pregnancies delivered after 37 weeks of gestation. The lack of significant associations of CRP and IL-1 with diet may be explained with the timing of blood collection and inundated by the natural spike of these mediators at parturition reported in term pregnancies, which may have obscured the signal of the prenatal nutrients [91,92]. The longitudinal assessment of blood specimens throughout pregnancy may increase the value of these biomarkers as predictors of diet-associated inflammation.

The identification for modifiable pregnancy risk factors for inflammation in utero is critical given that between 24–44 weeks gestation the brain undergoes its most rapid phase of growth, including cell proliferation, differentiation, synapse formation and myelination [93]. During this critical “growth spurt,” the brain is highly susceptible to the effects of inflammation, that activates microglia, leading to oligodendrocyte cell death [94,95], and white matter injury (WMI) [96,97]. For example, maternal and neonatal IL-8 elevation has been associated with ventriculomegaly, microcephaly [31], cerebral palsy [98], and neurodevelopmental delays in childhood-adulthood [32,99], and schizophrenia [100]. Thus, prenatal interventions targeting infection and improved nutritional status will have not only benefits for the mother, but potential long-term benefits for the offspring and human capital. Perumal and colleagues recently estimated that the scale up of provision of multiple micronutrients in pregnancy to 90% in LMICs would result in 5.02 million years of increased schooling and USD 18.1 billion in lifetime earnings, per birth cohort. This impact was modeled via the reduction in adverse birth outcomes (small for gestational age or preterm birth); inflammation is likely one of the key pathways by which these effects on neurodevelopment are mediated.

There are several limitations to our study. The data collected on pregnancy infections was based on recall and we did not have clinical data or medical records, although delivery data was also collected from birth attendant interview. Rates of self-reported antenatal infections were generally lower in our cohort than in other reports from Bangladesh or the US [101]. Dietary intake data was based upon FFQ, and we did not have portion size available. However, we had longitudinal repeated data collection to better estimate habitual intake [102], and used data from local Bangladeshi recommended serving size, a method that has been used in other studies [103]. While FFQ may not as accurately estimate absolute nutrient intake, it has high validity for ranking levels of dietary intake within populations [104]. Intake of several nutrients were highly correlated (r > 0.80) because intake values calculated were primarily based upon the food item frequency, and certain food items tended to be rich in several vitamins together (such as B vitamins). This resulted in similar inflammation risk effect sizes for correlated nutrients. Finally, not all nutrients were available in Bangladesh food tables, and for those with missing data we utilized Indian food composition tables that share common food items and ingredients.

## 5. Conclusions

In Sylhet, Bangladesh, a population where maternal undernutrition is prevalent, we found that low maternal dietary intake of several micronutrients in pregnancy (B vitamins, fat-soluble vitamins, zinc, iron, and linoleic acid) was significantly associated with risk of cord blood inflammation at the end of pregnancy. This data supports the hypothesis that targeting multiple micronutrient status via supplementation, may impact maternal-fetal inflammation responses. Maternal and fetal-neonatal inflammation are key mediators of adverse birth outcomes and offspring neurodevelopment [105]. In future analyses we plan to examine these relationships in sub-groups of vulnerable newborns (preterm, small for gestational age), and assess the associations of perinatal inflammation and longer-term neurodevelopmental outcomes and growth at 2 years. Identifying prenatal intervention targets to reduce in utero exposure to inflammation during the most sensitive and critical period of early brain development is a key strategy to optimize offspring neurodevelopment and human capital globally.

## Figures and Tables

**Figure 1 nutrients-13-03792-f001:**
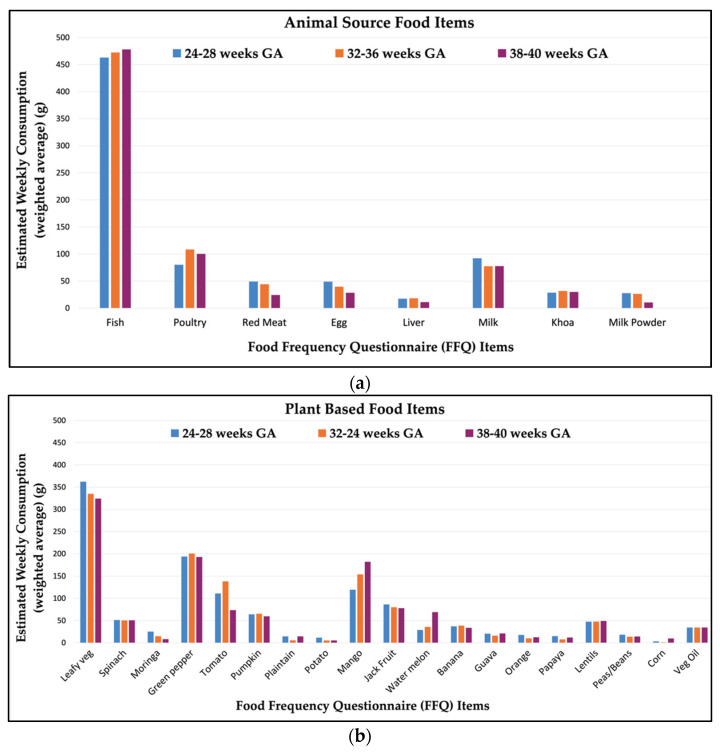
Estimated average weekly consumption (grams) of food items through the course of gestation among (*n* = 251) pregnant women in Sylhet, Bangladesh: (**a**) animal source and (**b**) plant-based food items.

**Figure 2 nutrients-13-03792-f002:**
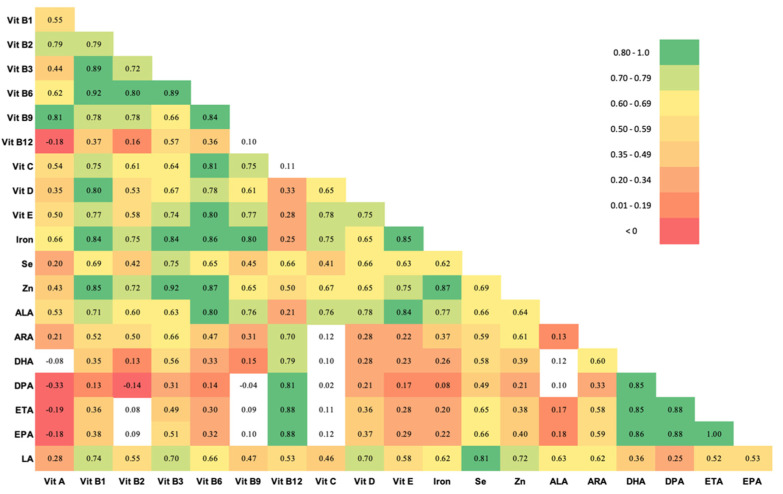
Spearman correlations between 20 nutrients in pregnant women in the Projahnmo cohort, Sylhet, Bangladesh. Unshaded cells did not meet statistical significance. Abbreviations: Vit = Vitamin, Se, selenium; Zn, Zinc; ALA, Alpha Linolenic acid; ARA, arachidonic acid; DHA, docosahexaenoic acid; DPA, docosapentaenoic acid; ETA, eicosatrienoic acid; EPA, eicosapentaenoic acid; LA, linoleic acid.

**Figure 3 nutrients-13-03792-f003:**
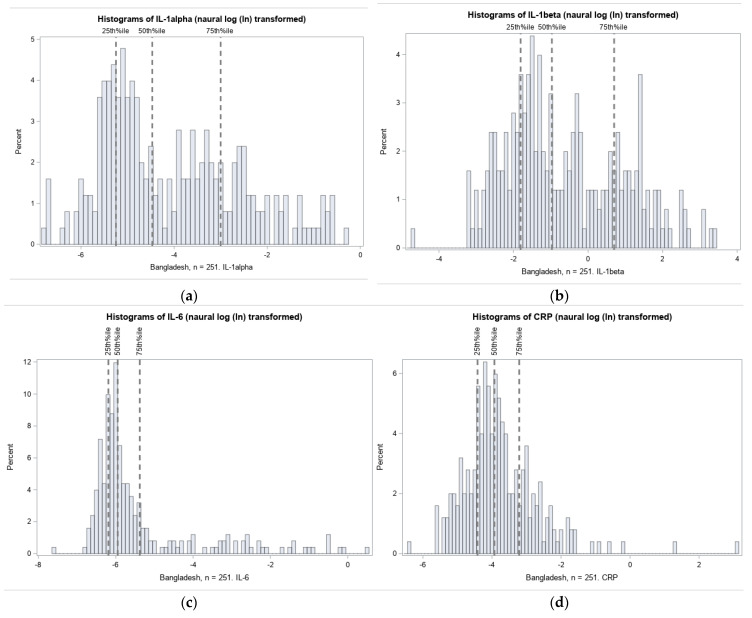
Distribution of inflammation biomarkers (natural log transformed) in Projahnmo cohort. The 75% thresholds for each cytokine are: (**a**) IL-1α: 0.05 pg/mg total protein; (**b**) IL-1β: 2.01 pg/mg total protein; (**c**) IL-6: 0.0046 pg/mg total protein; (**d**) IL-8: 3.09 pg/mg total protein; and (**e**) CRP: 0.04 ng/mg total protein.

**Figure 4 nutrients-13-03792-f004:**
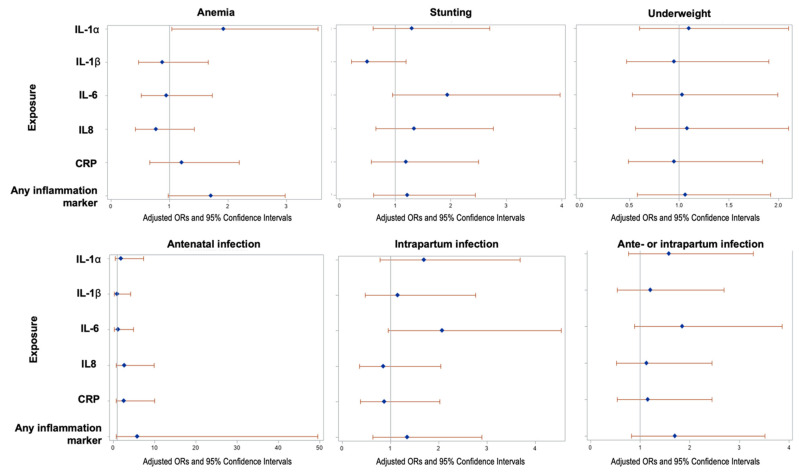
Association of maternal anthropometry and infections with newborn inflammation at delivery (*n* = 251 healthy term infants): Odds ratios are for outcome of elevation in inflammation protein above the 75% (vs. <75%). The 75% thresholds for each cytokine are: IL-1α: 0.05 pg/mg total protein; IL-1β: 2.01 pg/mg total protein; IL-6: 0.0046 pg/mg total protein; IL-8: 3.09 pg/mg total protein; and CRP: 0.04 ng/mg total protein. Analyses were adjusted for potential confounding by socioeconomic status, nulliparity, maternal education attainment (years), maternal upper arm circumference (MUAC) < 22 cm, tobacco or betel nut use, and season of initial antenatal assessment.

**Table 1 nutrients-13-03792-t001:** Basic characteristics of Projahnmo study population (*N* = 251) ^1^.

Maternal Characteristics	Mean ± SD or *N* (%)
Gestational weeks at enrollment (by ultrasound dating < 20 weeks) ^2^	12.1 ± 3.8
Age (years)	23.7 ± 4.6
Education (years) ^2^	6.0 ± 2.9
Wealth index ^3^	−0.04 ± 1.74
Parity	
Nulliparous	79 (31.5)
1–2	119 (47.4)
3+	53 (21.1)
Season during 24–28 week antenatal visit	
Grisma (summer, mid-April to mid June)	115 (45.8)
Barsa (rainy, mid-June to mid-August)	43 (17.1)
Sarat (autumn, mid-August to mid-October)	39 (15.5)
Hermanta (late autumn, mid October to mid-December)	1 (0.4)
Shhit (winter, mid-December to mid-February)	1 (0.4)
Basanta (spring, mid-February to mid-April)	52 (20.7)
Stunting (height < 145 cm) at enrollment ^2^ (cm)	50 (20.1)
BMI at enrollment ^2^	
Underweight (BMI < 18.5 kg/m^2^)	79 (31.7)
Normal (18.5 < BMI < 25 kg/m^2^)	152 (61.0)
Overweight (25 < BMI < 30 kg/m^2^)	18 (7.2)
Mid-upper arm circumference < 22 cm at enrollment ^3^ (cm)	85 (34.5)
Weight change from enrollment to 38–40 weeks GA3 (kg)	5.6 (4.0)
Weight change by week (kg/week)	0.23 ± 0.17
Maternal hemoglobin at enrollment ^2^ (g/dL)	11 (1.2)
Anemic (Hg < 11 g/dL)	117 (47.4)
Betel nut/tobacco use (chewing/sniffing during this pregnancy) ^4^	91 (36.3)
**Delivery and Infant Characteristics**	
Delivery at a health facility	230 (92.0)
Gestational age at birth ^2^ (by ultrasound dating < 20 weeks)	39.48 ± 1.2
Infant sex (female)	134 (53.4)
Birth weight ^2^ (g)	2749.9 ± 413.7
Birth weight z-score	−1.09 ± 1.01
Low birth weight ^2^ (<2500 g)	51 (22.9%)
Infant Size for Gestational Age ^2^	
Small for gestational age (SGA) (birthweight z-score < 10 percentile)	97 (43.9)
Appropriate for gestational age (AGA) 10th–90th percentile)	122 (55.2)
Large for gestational age (LGA) (>90th percentile)	2 (0.9)

^1^ Excluded stillbirths, infants who died shortly after delivery, neonatal encephalopathy, and preterm (GA < 37 weeks). All dyads are singleton births. ^2^ Missing values (*n* (%)): Maternal characteristics: education = 1 (0.4); height = 2 (0.8); BMI = 2 (0.8); MUAC = 5 (2.0); weight change = 4 (1.6); hemoglobin = 4 (1.6); gestational age = 1 (0.4). Infant characteristics: birth weight = 28 (11.2); birth weight z-score = 30 (12.0); low birth weight = 28 (11.2); Infant size for gestational age = 30 (12.0). ^3^ Household index was created from wealth scores based on housing materials and household possessions using principal component analysis [44,45]. ^4^
*n* = 91 women reported using betel nut and *n* = 54 women reported tobacco use; all women using tobacco were also using betel nut (59.3% of *n* = 91 reporting both betel nut and tobacco use).

**Table 2 nutrients-13-03792-t002:** Daily nutrient intake during Pregnancy in Projahnmo Cohort, Sylhet, Bangladesh (*n* = 244).

Nutrient	Mean (SD)	Median (IQR)	Range	IOM Recommended Dietary Allowances * and Adequate Intakes ^†^
Vitamin				
Vitamin A (µg)	458 (357)	380 (241–551)	32.05–2741	770 *
Vitamin B1 (mg)	0.28 (0.1)	0.27 (0.21–0.35)	0.09–0.56	1.4 *
Vitamin B2 (mg)	0.47 (0.2)	0.45 (0.33–0.60)	0.12–1.60	1.4 *
Vitamin B3 (mg)	8.82 (2.6)	8.29 (6.8–10.7)	3.55–16.62	18 *
Vitamin B6 (mg)	0.59 (0.2)	0.56 (0.42–0.74)	0.20–1.49	2.0 *
Vitamin B9 (Folate) (µg)	104.80 (54)	95.2 (67.0–133)	9.36–321	600 *
Vitamin B12 (µg)	1.67 (0.5)	1.77 (1.42–1.97)	0.20–2.75	2.6 *
Vitamin C (mg)	84.70 (46)	77.6 (48.8–115)	1.50–261	80–85 *
Vitamin D (µg)	10.08 (4.4)	9.43 (6.60–12.6)	2.26–27.1	15 *
Vitamin E (mg)	3.16 (0.8)	3.06 (2.50–3.69)	1.76–6.01	15 *
Mineral				
Iron (mg)	3.57 (1.3)	3.47 (2.5–4.5)	1.05–7.14	27 *
Selenium (mg)	0.04 (0.01)	0.04 (0.03–0.05)	0.01–0.08	0.06 *
Zinc (mg)	3.22 (1.2)	3.02 (2.3–4.0)	1.07–7.60	11–12 *
Omega-3 LCPUFA				
Alpha Linolenic acid (mg)	201.77 (107.4)	188.8 (111–263)	20.0–575	1400 ^†^
Docosahexaenoic acid (mg)	75.54 (16.1)	83.0 (68.6–85.8)	17.65–118	-
Docosapentaenoic acid (mg)	32.35 (9.4)	34.6 (30.4–38.6)	2.77–38.7	-
Eicosapentaenoic acid (mg)	26.39 (7.64)	28.22 (24.8–31.6)	2.26–31.6	-
Eicosatrienoic acid (mg)	21.68 (6.4)	23.23 (19.4–25.6)	3.27–35.7	-
Omega-6 LCPUFA				
Linoleic acid (g)	588 (305)	511 (390–712)	180–2270	13 ^†^
Arachidonic acid (mg)	87.41 (34)	79.1 (62.2–105)	33.2–325	-

*,† Values for Recommended Dietary Allowances and Adequate Intakes were calculated and reported by the Institute of Medicine (IOM) Food and Nutrition Board for pregnancy ages 14–50 years [46]. An RDA * is the average daily dietary intake level sufficient to meet the nutrient requirements of nearly all (97–98 percent) healthy individuals in a group. An AI† is believed to cover the needs of all healthy individuals in the groups, but lack of data prevents being able to specify with confidence the percentage of individuals covered by this intake. (-) Denotes unavailable data.

**Table 3 nutrients-13-03792-t003:** The adjusted odds of cord blood inflammation (≥75th percentile) by tertile of maternal nutrient intake (*n* = 251 mothers and their term infants). (Reference is the middle tertile of nutrient intake).

Nutrient ^1^	Tertiles of Intake (Range)	IL-1α	IL-1β	IL-6	IL-8	CRP	Any Marker ^2^
		aOR ^1^ (95%CI)	aOR ^1^ (95%CI)	aOR ^1^ (95%CI)	aOR ^1^ (95%CI)	aOR ^1^ (95%CI)	aOR ^1^ (95%CI)
Vit A (mcg)	<33% (32.1–286)	1.50 (0.7–3.21)	2.13 (0.93–4.87)	1.18 (0.56–2.52)	1.76 (0.81–3.79)	1.16 (0.56–2.41)	1.27 (0.64–2.51)
	>67% (482–2741)	0.96 (0.44–2.1)	1.81 (0.8–4.1)	0.78 (0.36–1.69)	0.79 (0.35–1.78)	0.62 (0.29–1.35)	1.15 (0.58–2.28)
Vit B1 (mg)	<33% 0.09–0.22)	2.13 (0.98–4.65)	1.76 (0.79–3.95)	2.07 (0.95–4.48)	**2.27 (1.05–4.91) ***	1.11 (0.52–2.34)	1.14 (0.57–2.27)
	>67% (0.32–0.56)	1.30 (0.59–2.85)	0.75 (0.34–1.68)	1.02 (0.46–2.26)	0.74 (0.32–1.67)	1.01 (0.48–2.12)	0.91 (0.46–1.8)
Vit B2 (mg)	<33% (0.12–0.37)	**3.17 (1.39–7.19) ***	2.11 (0.93–4.8)	**2.25 (1.03–4.94) ***	**2.45 (1.12–5.37) ***	**2.52 (1.17–5.45) ***	**2.84 (1.37–5.88) ***
	>67% (0.55–1.60)	1.86 (0.83–4.15)	1.14 (0.52–2.51)	1.10 (0.5–2.41)	0.74 (0.33–1.68)	1.10 (0.5–2.42)	1.45 (0.74–2.86)
Vit B3 (mg)	<33% (3.55–7.34)	**2.30 (1.03–5.12) ***	**2.94 (1.25–6.9) ***	1.60 (0.74–3.46)	**3.23 (1.42–7.31) ***	0.96 (0.46–2.02)	1.70 (0.84–3.44)
	>67% (0.20–0.56)	1.61 (0.72–3.58)	1.70 (0.75–3.84)	0.80 (0.36–1.78)	1.32 (0.57–3.06)	0.62 (0.29–1.34)	1.04 (0.52–2.06)
Vit B6 (mg)	<33% (0.20–0.47)	1.78 (0.82–3.87)	1.92 (0.85–4.34)	**2.47 (1.12–5.44) ***	**2.35 (1.08–5.13) ***	1.56 (0.74–3.3)	1.43 (0.71–2.9)
	>67% (0.68–1.49)	1.20 (0.55–2.61)	0.75 (0.34–1.67)	1.09 (0.49–2.44)	0.71 (0.31–1.63)	0.90 (0.41–1.95)	0.80 (0.4–1.58)
Vit B9 (Folate) (mcg)	<33% (9.36–77.3)	1.60 (0.73–3.51)	2.04 (0.88–4.71)	1.16 (0.54–2.47)	**2.28 (1.04–5.02)***	1.36 (0.64–2.89)	1.14 (0.57–2.3)
	>67% (121–321)	1.28 (0.6–2.73)	1.26 (0.57–2.76)	0.55 (0.25–1.19)	0.78 (0.35–1.76)	0.86 (0.4–1.85)	0.89 (0.45–1.76)
Vit B12 (mcg)	<33% (0.20–1.60)	0.60 (0.28–1.3)	0.49 (0.21–1.13)	**0.42 (0.19–0.92) ***	0.49 (0.23–1.07)	0.75 (0.35–1.6)	0.83 (0.41–1.67)
	>67% (1.89–2.75)	0.63 (0.29–1.37)	0.54 (0.24–1.24)	**0.38 (0.17–0.84) ***	**0.42 (0.19–0.95) ***	0.71 (0.33–1.56)	0.70 (0.35–1.4)
Vit C (mg)	<33% (1.50–57.3)	1.28 (0.59–2.8)	1.28 (0.57–2.89)	1.31 (0.61–2.81)	1.23 (0.57–2.62)	0.65 (0.31–1.38)	0.89 (0.45–1.77)
	>67% (101–261)	1.18 (0.56–2.48)	0.76 (0.35–1.65)	0.82 (0.38–1.74)	0.52 (0.24–1.15)	0.52 (0.25–1.1)	0.88 (0.45–1.73)
Vit D (mcg)	<33% (2.26–7.71)	0.66 (0.29–1.52)	1.44 (0.59–3.52)	1.33 (0.57–3.07)	1.18 (0.51–2.73)	0.90 (0.39–2.06)	0.84 (0.39–1.79)
	>67% (11.6–27.13)	**0.48 (0.23–0.99) ***	0.65 (0.31–1.37)	0.80 (0.39–1.67)	0.57 (0.27–1.2)	0.86 (0.42–1.77)	**0.46 (0.23–0.91) ***
Vit E (mg)	<33% (1.76–2.74)	0.91 (0.42–1.98)	1.00 (0.43–2.34)	0.82 (0.38–1.79)	1.06 (0.49–2.3)	0.90 (0.43–1.9)	0.54 (0.27–1.09)
	>67% (3.45–6.01)	0.79 (0.38–1.65)	0.76 (0.36–1.61)	0.60 (0.28–1.26)	**0.42 (0.19–0.92) ***	0.56 (0.26–1.19)	**0.44 (0.22–0.89) ***
Iron (mg)	<33% (1.05–2.76)	1.57 (0.73–3.38)	**2.37 (1.03–5.45) ***	1.52 (0.72–3.25)	**2.28 (1.05–4.97) ***	0.94 (0.45–1.94)	1.06 (0.54–2.09)
	>67% (4.20–7.14)	1.30 (0.61–2.78)	1.52 (0.68–3.37)	0.96 (0.45–2.05)	1.01 (0.45–2.24)	0.70 (0.34–1.48)	0.75 (0.38–1.46)
Selenium (mg)	<33% (0.01–0.03)	1.43 (0.67–3.06)	1.62 (0.72–3.66)	1.01 (0.48–2.16)	1.79 (0.84–3.83)	0.95 (0.46–1.97)	0.95 (0.48–1.89)
	>67% (0.04–0.08)	1.23 (0.58–2.62)	1.43 (0.66–3.1)	1.01 (0.48–2.11)	1.02 (0.47–2.23)	0.65 (0.3–1.37)	0.66 (0.33–1.3)
Zinc (mg)	<33% (1.07–2.56)	2.10 (0.95–4.65)	**2.52 (1.07–5.91) ***	2.08 (0.93–4.65)	2.20 (1–4.86)	1.60 (0.73–3.47)	1.75 (0.86–3.58)
	>67% (3.61–7.60)	1.23 (0.56–2.7)	1.26 (0.56–2.82)	1.24 (0.57–2.73)	0.68 (0.3–1.55)	1.06 (0.49–2.28)	0.97 (0.49–1.92)
Linoleic acid (mg) ^3^	<33% (1020–1994)	1.72 (0.81–3.67)	1.73 (0.76–3.9)	1.35 (0.63–2.9)	**2.35 (1.07–5.15) ***	1.47 (0.69–3.12)	1.89 (0.93–3.86)
	>67% (2116–3561)	0.78 (0.36–1.69)	1.10 (0.51–2.38)	0.85 (0.4–1.8)	0.96 (0.43–2.12)	0.87 (0.41–1.85)	0.83 (0.42–1.64)
**Low Nutrient Intake Categories ^4^**							
Water-soluble vitamins ^5^		1.08 (0.55–2.13)	1.28 (0.64–2.57)	1.19 (0.6–2.35)	1.60 (0.79–3.24)	1.40 (0.71–2.79)	1.07 (0.58–1.97)
Fat-soluble vitamins ^5^		1.33 (0.7–2.55)	1.68 (0.86–3.27)	1.18 (0.61–2.26)	**1.96 (1.01–3.82) ***	1.55 (0.81–2.95)	1.34 (0.74–2.43)
Minerals		1.51 (0.8–2.85)	**2.12 (1.07–4.20) ***	1.23 (0.65–2.32)	**2.07 (1.07–3.99) ***	1.68 (0.9–3.15)	1.49 (0.84–2.65)
LCPUFAs		1.03 (0.53–1.98)	0.90 (0.45–1.80)	1.99 (0.98–4.04)	1.42 (0.72–2.83)	1.38 (0.70–2.71)	0.96 (0.53–1.75)

^1^ Reference category for models of individual nutrient intake tertiles are levels between the 33rd–67th percentile of the sample distribution. Models adjusted for socioeconomic status (continuous), primiparity, maternal upper-arm circumference < 22 cm at enrollment, tobacco and/or betel nut use, season of initial antenatal assessment, and maternal educational attainment (years). Analytic sample, *n* = 244 dyads with both nutrient and inflammatory marker data available. ^2^ Any marker refers to elevation (≥75th percentile) of one or more of the five inflammatory markers examined. ^3^ Adjusted odds ratios of LCPUFAs not present in this table were not statistically significant, effect estimates are provided in Appendix A. ^4^ Low nutrient intake category variables are binary: operationalized as one or more deficiencies within nutrient category (at least one nutrient with value < 33rd percentile) versus no deficiencies (reference, no values less than 33rd percentile) within nutrient category. ^5^ Water-soluble vitamins include all Bs and C; fat-soluble vitamins include A, D, and E. * Statistically significant odds ratios (*p* < 0.05) are bolded.

## Data Availability

Data may be made available upon request and approval by the Projahnmo research committee and with regulatory permissions.

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
