# Peer review of "Maternal Diet, Infection, and Risk of Cord Blood Inflammation in the Bangladesh Projahnmo Pregnancy Cohort"

_nutrients, 2021, doi:10.3390/nu13113792_

Round 1
Reviewer 1 Report
Major:
- Line 167-182: Define data presentation uniformly as medians and IQR, where important the full range, too, unless ALL data are normally distributed. The study population appears to be small. The authors should provide a power calculation or describe, why there is none.
- Data should be trichotomized in <25%, 25-75% and >75% of values to better see the extremes.
- Moreover, individual parameters easily can be categorised (e.g.: deficiency of 1 to x items in a group of related parameters to provide an idea of single or multiple micronutrient deficiencies in relation to a group of inflammatory/infectios events. This may improve the - so far not existing - statistical value of data, and their correlation/regression with clinical items.
- Fig 1: The figure isnot well designed as the fonts are much too small to be read. Better separate into A,B C and sub-groups of food sources (animal-derived, vetetables, fruits). Estimated amounts according to your protocol of ingested food should be indicated rather than only the frequency alone.
- Table 2: Median and IQR are generally more adequate than means/SD, particularly due to the statistical approach of lower/upper quantile analysis. Why is vitamin B2 is missing? Concerning iron: Is there clinical data on anemia of the pregnant or hematocrit of the fetus (cord blood was measured!)
- Table 2, legend, reference 42: Importantly, replace the source and values of RDA/AI recommendations by an adequate organisation's reference. The Bill and Melinde Gates Foundation Report is not a peer-reviewed citation.
- Fig 2: Similar to Fig. 1, the font size of figure is much too small!. If it were not square-saped, this would be easier. Indicate colour meanings, and ALL abbreviations, please.
- line 271: May be, the authors wish to put this important aspect into the main document.
- line 284: This actually means that either there is no evidence for your conclusions or the way you calculated/presented the data is wrong.
- Fig. 3: This way of documentation isn't an association, particularly as there is not a single signifucant odds ratio. Try to sum up parameter values or use a way of indicating inflammatory situations like "increase of 1 or more markers, and correlate it with either single or multiple klinical parameters (Also see 3.). This may improve and simplify the read-out of the huge amount of data, and may improve the actually not existing clinical significance and impact.
- Table 3: Provide explanation of this calcuklation mode.
Provide international ref values of a peer-reviewed source. Riboflavin is missing. - Line 311, Discussion: The discussion is a recitation of mediator function (partly belonging to the introduction) rather than a discussion of observations. Otherwise, in such a review-like discussion of micronutrients the mechanisms how they contribute to mediator synthesis, must be provided.
- Line 316f: According to your data presentation you DID NOT confirm this statement.
- Line 341f: Where is the table or figure showing this in your study.
Minor:
- line 63: remove 'of'
- line 193f and Table 1: Does the basis of underweight estimation match with the population of Bangladesh?
- Table 1, Infant Size for Gestational Age: Provide unit. Is 2500g adequate for Bangladeshi?
- line 260: Put antenatal infection rate in relation to the general frequency of infections, in non-pregant Bangladeshi women and West-European/USA pregnancies.
- line 377: Provide reference, please.
- line 399: Remove 'it'.
Reviewer 2 Report
This article is very interesting, the participation of the nutritional behavior of the pregnant woman and its association with inflammatory cytokines at delivery is of great interest to know important aspects such as factors that may precipitate neonatal sepsis or chorioamnionitis. It is an article in a vulnerable population with nutritional deficit during gestation, which in my opinion gives more value to the data. Nevertheless, I would like to share some comments with the authors to improve this work and discuss with them other points of view.
In the introduction, after reading the article, there is no mention of aspects of neonatal neurodevelopment (lines 74-75), nor with prematurity, which also appears at the end of the introduction (lines 76-77), so perhaps they could be deleted or reordered.
Regarding the material and methods. I was very surprised and low that the economic level was approximately $60/month. On the other hand, it is not clear to me the inclusion criteria of the sample, nor if from the 299 women, 48 were excluded to have a final number of 251 on which to make the subsequent analysis.
Could the BMI around 12 weeks of gestation be considered as a pre-gestational index?
At the 38-40 week point there were no anthropometric measurements?
In my opinion, it is a work with many points and samples, I think it would make it much easier to have a figure/table summarizing the collection points and what data were recorded in each point.
I would like to congratulate for the inflammatory profile analysis, although the MSD protocol could be minimally detailed, in general it is very understandable.
In the statistical part, it is not clear to me the justification of categorizing the inflammatory variables, would not it have more powerful to introduce the variables in the models in a quantitative way? Similarly, the article described correlations that are not mentioned in this section.
In the results section, Figure 1 should be larger or split in two, as it is difficult to read the axes and the categorization of the weeks of gestation. In addition, the title accompanying the figure should be placed at the bottom.
My concern is the correlogram, what does it add to the interest of the article? what do you want to achieve with this information? in fact, it does not show which correlation is significant and which is not. It seems logical to think that the micronutrients correlate with each other, but this does not indicate any kind of causality between them. Furthermore, being a matrix, the upper quadrant is equal to the lower one, to make the matrix more understandable, I suggest that, if the authors consider keeping this figure, only one of the quadrants should be shown, with the non-significant cells blank.
In the section on maternal nutrition, infection and risk of inflammation, the question of the categorized cytokine measures comes back to me.
Also, on lines 262-263, it mentions that urinary tract infection was the most reported, however it is an n=2, while for treating an infection during pregnancy, it has an n=10. I think it should be clarified.
Discussion. It is very interesting to read, and I think that in general, it is very well put together. The authors mention a dose dependence (line 319), this was because of the categorization they apply? I don't know if in this case it can be considered dose-dependence or by segments.
Considering these data and the literature they mention in studies with other cohorts in high-income countries, we could assume that both a nutritional deficit or excess during gestation is associated with an imbalance in the inflammatory profile at the end of pregnancy, which could be associated with maternal-neonatal inflammatory complications. In short, a balanced maternal nutrition, and therefore its follow-up by health professionals, should be the guideline during gestation.
In the conclusions, I would like to note that I do not know if it is correct to mention that they have identified neonatal risk factors for inflammation at birth (line 433-434), since the study reports prevalences and there is no subsequent neonatal follow-up. Perhaps, that elevated inflammatory profile is beneficial in the long-term. Which leads me to question the title of the article, as I do not believe that "infection and risk of neonatal inflammation" are two outcomes of this work.
Instead, it should be added in the conclusions that the intake of micronutrients throughout gestation was associated with a decrease in two potentially pro-inflammatory cytokines, IL-6 and IL-8, at the end of pregnancy. Evidence that nutritional intervention during pregnancy should be continued and that it is associated, at least moderately, with changes in the inflammatory system.
Minor comments:
- line 77, IQ is not defined above.
- line 121, ANC is not defined above.
- line 123, MUAC is not defined above.
- line 153, behind the hospital include "(Boston, MA, USA)".
- line 182, "That" is in italics
- I could not see the whealth index of table 1 in material and methos
- line 245, "LC-Pufa's" should write "LCPUFAs"
- line 297, "table 3" is bold
